



# A new method for the pragmatic choice of wind models for Wind Resource Assessment in complex terrain

Sarah Barber[1], Alain Schubiger[1], Natalie Wagenbrenner[2], Nicolas Fatras[3], and Henrik Nordborg[1]

[1]University of Applied Sciences Rapperswil, Oberseestrasse 10, 8640 Rapperswil, Switzerland
[2]Missoula Fire Sciences Laboratory, 5775 W. US Hwy 10, Missoula, MT 59808, USA
[3]Zephy Science, Lubeckerstr. 139, 22087 Hamburg, Germany

**Correspondence:** Sarah Barber (sarah.barber@hsr.ch)

**Abstract.** The accuracy of the estimation of the wind resource has an enormous effect on the expected rate of return of a wind energy project. Due to the complex nature of the weather and the wind flow over the earth's surface, it can be very challenging to measure and model the wind resource correctly. For a given project, the modeller is faced with a difficult choice of a wide range of simulation tools with varying accuracies (or *skill*) and costs. In this work, a new method for helping wind modellers choose

the most cost-effective model for a given project is developed by applying six different Computational Fluid Dynamics tools to simulate the Bolund Hill experiment and studying appropriate comparison metrics in detail. This is done by firstly defining various parameters for predicting the *skill* and *cost* scores **before** carrying out the simulations as well as for calculating *skill* and *cost* scores **after** carrying out the simulations. Weightings are then defined for these parameters, and values assigned to them for the six tools using a template containing pre-defined limits in a blind test. An iterative improvement process is applied

by collecting inputs from the participants of the study. This allows a graph of predicted *skill* score against *cost* score to be produced, enabling modellers to choose the most cost-effective model without having to carry out the simulations beforehand. The most effective model is the one with the highest *skill* score for the lowest *cost* score, at the flattening-off part of the curve. The results show that this new method is successful, and that it is generally possible to apply it in order to choose the most appropriate model for a given project in advance. This is demonstrated by the good match between the shapes of the *skill* score

against *cost* score curves **before** and **after** the simulations, and by the fact that the tool at the flattening-out point of the curve is the same **before** and **after** carrying out the simulations. It is also shown how important it is to take into account other factors that may affect the accuracy and costs of a wind modelling simulation as well as the quality of the aerodynamic equations and the run-time. Several improvements to the method are being worked on, by further examining the discrepancies between the predicted and actual *cost* and *skill* scores. Additionally, the method is being extended for calculating all wind directions and

the Annual Energy Production, as well as to include mesoscale nesting or forcing. A large number of inputs are being collected as part of a simulation challenge in collaboration with IEA Wind Task 31. The method has a high potential to be extended to a wide range of other simulation applications.





# 1 Introduction

In wind energy, the accuracy of the estimation of the wind resource has an enormous effect on the expected rate of return of a project. Due to the complex nature of the weather and of the wind flow over the earth's surface, it can be very challenging to measure and model the wind resource correctly. For a given project, the modeller is faced with a difficult choice of a wide

range of simulation tools with varying accuracies and costs. Additionally, different tools have different functionalities - some calculate the entire wind climate (all wind directions) and the energy production, whereas some have to be manually set up to extract this information. Some include mesoscale nesting or forcing, whereas others focus only on microscale features. If the choice of model is made incorrectly, either many resources are wasted in needlessly high accuracy simulations, or the rate of return is inaccurate and investors risk losing large amounts of money. As there are currently no guidelines or tools available to

the modeller to help with this choice, it is usually left to gut feeling – and this can be catastrophic for investors or acquirers of wind farms.

In order to help modellers with this choice, a collaborative project has been started at the University of Applied Sciences Rapperswil, together with Meteotest AG, Hochschule Esslingen and Stadtwerke Tubingen. In this project, a decision tool is being developed to help modellers choose the most appropriate model for a particular wind energy related project. This involves

firstly applying various simulation tools with different fidelities, ranging from WAsP (Wind Atlas Analysis and Application Model), RANS-CFD (Reynolds-Averaged Navier-Stokes Computational Fluid Dynamics), DES (Detached Eddy Simulations), and LES (Large Eddy Simulations) to LBM (Lattice Boltzmann Method) to a range of test sites, defining appropriate comparison metrics and developing a draft decision process. Relevant comparison metrics include factors relating to the *skill* or accuracy of the model as well as those relating to its *cost*. In a second step, the simulation tools will be applied to a demonstra-

tion site in order to validate, demonstrate and improve the decision process.

This new decision process will allow modellers to choose the appropriate tool for a particular site by creating a plot similar to the one shown in Fig. 1. This shows a schematic representation of the *skill* score against *cost* score for a range of different tools, which are represented by the individual points. The areas marked in red are the areas deemed unacceptable by the modeller, where the *skill* score is too low and the *cost* is too high. These areas may vary depending on the expectations and requirements

of the modeller. The most effective solution is then chosen as the one with the highest *skill* score for the lowest *cost* score within the acceptable region, at the flattening-off part of the curve. Theoretically, it would be possible to estimate the *skill* and *cost* scores of each model based on the simulation run time and on the deviation of the results from measurements. However, this would not only require high quality measurement data to be available, it would require the modeller to run and compare a wide range of simulations with all the different models. This would completely defeat the point of the exercise - which is to choose

the most appropriate model **before** carrying out any simulations. The main focus of the project is therefore on developing transfer functions, which help modellers accurately estimate the *skill* and *cost* scores of different models for a given project **before** carrying out any simulations.





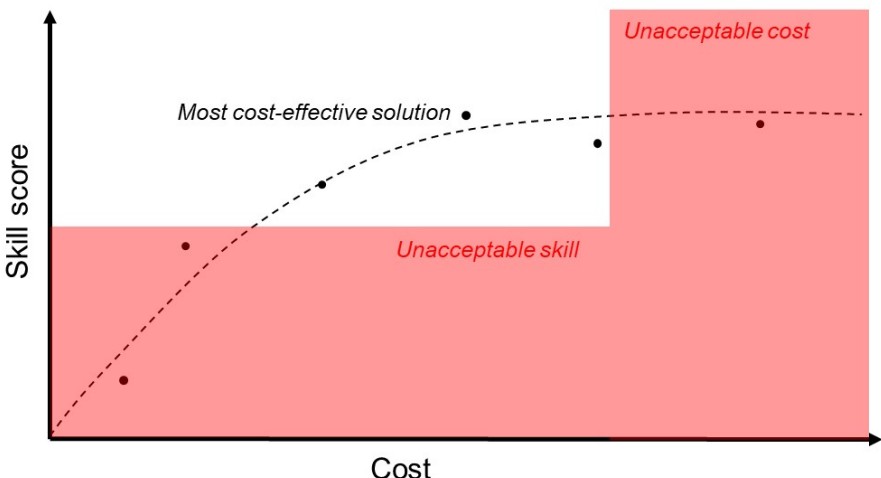

**Figure 1.** Expected skill score against costs for an example wind energy project.

This paper presents a new method for the pragmatic choice of wind models for Wind Resource Assessment, by studying appropriate comparison metrics for *skill* and *cost* scores using simulations carried out on the Bolund Hill experiment set-up (Berg et al., 2011), (Bechmann et al., 2011). The paper starts with a review of previous work regarding comparison metrics in Section 2, both in the area of CFD and in the area of wind energy, then introduces the method applied in Section 3, discusses the results in Section 4, and finishes with the conclusions in Section 5.

## 2 Previous work on comparison metrics

### 2.1 Computational Fluid Dynamics (CFD)

Work on the evaluation of the quality of CFD tools has previously been done in the COST Action 732 (Britter and Baklanov, 2007), with the objective of determining and improving model quality of microscale meteorological models for the prediction of flow and dispersion processes in urban and industrial environments. The work was based on the AIAA guideline for the verification and validation of CFD simulations (AIAA, 1998), which consists of a framework containing the three environments: reality, computerised model and conceptual model (which contains all the relevant equations). The phrases "model validation", "model qualification" and "model verification" refer to the difference between reality and the computerised model, the difference between reality and the conceptual model and to the difference between the conceptual and computerised models, respectively. The COST Action involved defining the following evaluation process: (1) Scientific evaluation process; (2) Verification process; (3) Provision of validation data sets; (4) Model validation process; (5) Operational evaluation process. As part of the model validation process, a range of different metrics were defined, including: correlation coefficient, Fractional Bias, Figure of Merit and Hit Rate. This can be evaluated using a statistical evaluation tool, comparing model predictions with



observations (reference states). These metrics can only be used if the number of data points is high enough to allow statistical analysis. For wind energy applications, ten-minute averages are usually sufficient, and therefore these metrics are not necessarily applicable here. However, some of the general ideas have been used to develop the method in this work.

Another relevant project on CFD evaluation was the SMEDIS Project (Daish et al., 2000), which involved developing a methodology for the scientific evaluation of dense gas dispersion models. A large part of this methodology involved a questionnaire that had to be filled out by the modellers, which asked them questions regarding pre-defined evaluation criteria. These included topics such as the purpose of the model as well as the physical and chemical processes modelled. A similar study involved the development of a guideline for the scientific evaluation CFD studies, focusing on factors such as the domain description and the grid set-up, the input data, the turbulence closure, the equation system and solver applied, the boundary

conditions, the initial conditions and the output data, as well as various parameterisations important for microscale modelling (VDI, 2005).

    All of these previous studies have been used as a basis for the development of the comparison metrics in this work, as described further in Section 3. In particular, the ideas mentioned above have been combined in order to develop a pre-defined template for participants to define individual comparison metrics.

## 2.2   Wind energy

There is no published work known to the authors that specifically compares the *skill* and *cost* of wind modelling tools for the wind energy industry. The New European Wind Atlas (NEWA) Meso-Micro Challenge for Wind Resource Assessment as part of the IEA Wind Task 31 "Wakebench" is the only related project known to the authors, and this aims to determine the applicability range of meso-micro methodologies for wind resource assessment within the NEWA validation domain (IEA,

2019). It does consider the relationship between tool accuracy and cost; however, no attempt is made to predict these parameters in advance in order to help modellers choose the best tool for a given project. Due to these synergies, the current work described in this paper is being carried out in collaboration with IEA Wind Task 31. Also as part of IEA Wind Task 31, a Wind Energy Model Evaluation Protocol (WEMEP) has been developed (Rodrigo, 2019). WEMEP addresses quality assurance of models being used for research and to drive wind energy applications. This is achieved through a framework to conduct formal

verification and validation (V&V) that ultimately determines how model credibility is built upon. It is based on the AIAA guide for the V&V of CFD as described in the previous section.

    Additionally, some metrics do exist for the definition of complex terrain, which may be relevant to the present work. The industry-standard linear wind field prediction tool WAsP (Wind Atlas Analysis and Application Model) is applicable for slopes up to 30% (Bowen and Mortensen, 1996), because it is generally recognised that flow separation is likely to occur above this

value (Wood, 1995). WAsP can be used to self-calculate the extent to which the terrain violates this requirement, using the Ruggedness Index (RIX), defined as the fractional extent of the surrounding terrain which is steeper than 30%. The orographic performance indicator $\Delta$RIX is defined as the difference in the (percentage) fractions between the predicted and reference sites. If the reference and predicted sites are equally rugged ($\Delta$RIX = 0%), the prediction errors are relatively small. As well as this, there have been attempts to assess flow complexity objectively using streamlines and other features e.g. Pozo et al. (2017).



However, these methods are sometimes computationally demanding or require functionality that is not available in wind energy sector tools.

These ideas have all been considered in the development of the comparison metrics in this work, as described in the next section.

## 3 Method

Comparison metrics for *cost* and *skill* scores were investigated using results of simulations from the Bolund Hill experiment (Berg et al., 2011), (Bechmann et al., 2011) using various wind modelling tools. This site was chosen due to the number and quality of measurement points as well as the complexity of the terrain. As this experiment was designed to exclude climatology effects, the present study is limited to microscale modelling of the wind. Parameters relating to geostrophic effects and mesoscale coupling or forcing will be investigated in a later study.

Bolund Hill is a natural hill that is 12 m high, 130 m long and 75 m wide, located in Roskilde Fjord, Denmark. It is surrounded by water in all directions except to the east. Ten-minute average wind speed data is available from a total of 38 measurement locations on nine different meteorological masts for different wind directions. A contour map of the hill with the meteorological masts marked is shown in Fig. 2. The blind test experiment conducted in 2009 consisted of the simulation of four wind direction cases (270º, 255º, 239º and 90º) with prescribed boundary conditions of neutral flow (Bechmann et al., 2011). A more detailed description can be found in the literature, e.g. Bechmann et al. (2011), Berg et al. (2011).

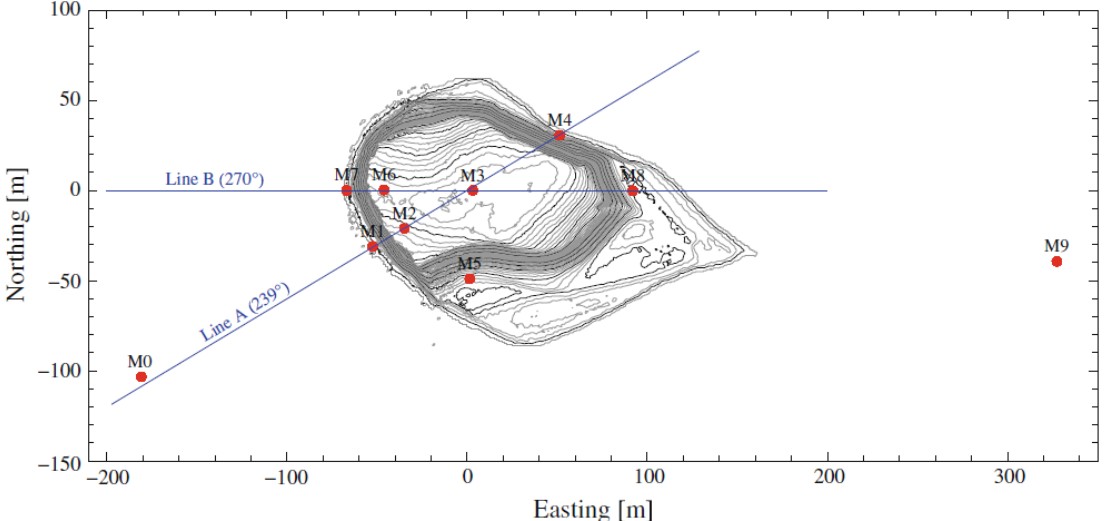

**Figure 2.** Bolund Hill contour map with 0.25 m contour interval.

The method used in this work in order to study the applicability of comparison metrics for wind modelling in complex terrain is shown in Fig. 3. It involved: (1) Definition of the test case; (2) Definition of parameters for comparison metrics before





and after running simulations; (3) Definition of weightings for each parameter; (4) Scoring of each parameter for each model applied; (5) Calculation of weighted score of each parameter for each model applied; (6) Plot and comparison of skill scores vs. cost scores before and after running simulations; (7) Sensitivity studies. After step (6), the chosen weightings were improved iteratively by comparing the total scores with expected values. The ultimate goal is to use the results in order to develop a

5    method capable of estimating the *skill* and *cost* scores of a given model for a given project without having to carry out the simulations beforehand.

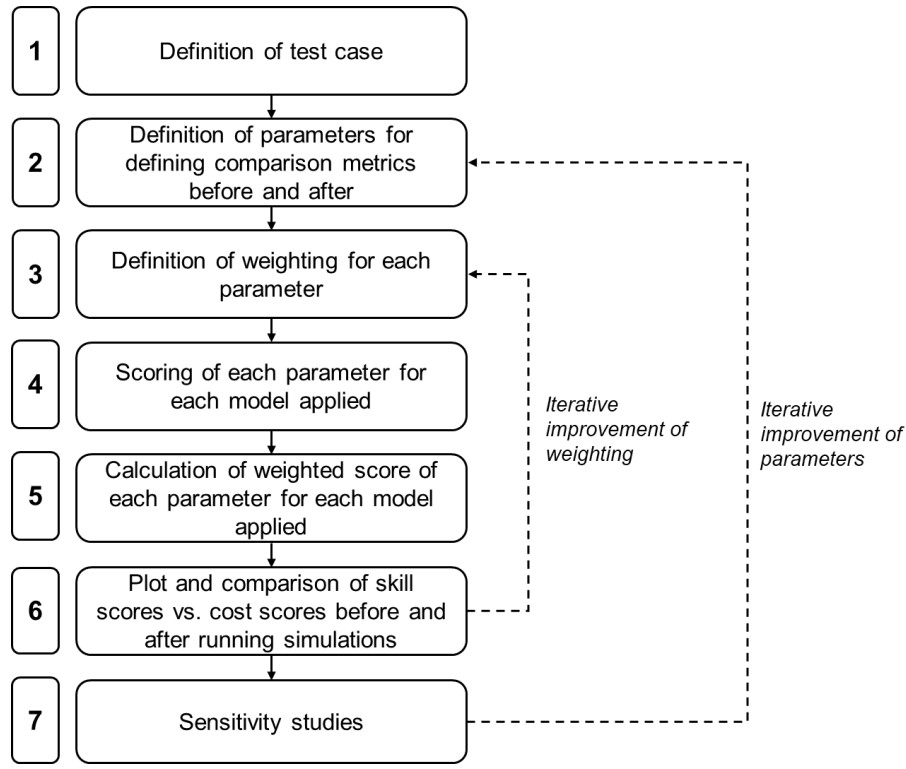

**Figure 3.** Flow diagram of the method applied in this work.

The wind modelling tools that were applied in this work are described below. For each tool, the boundary conditions were set up as required in the Bolund Hill blind test, defined by the average wind speed profile and the turbulence intensity at met mast number M0 (Bechmann et al., 2011).

10   **3.1    Model 1: *WindNinja-COM***

*WindNinja* is a microscale wind modelling framework developed for operational wildland fire applications (Forthofer et al., 2014a), (Forthofer et al., 2014b). It is specifically designed to meet the needs of emergency response personnel including simple inputs, fast simulation times on simple hardware, and minimal training requirements. To this end, *WindNinja* includes an easy-to-use graphical user interface with flexible initialisation options, a function for downloading the data required for



model initialisation, built-in tutorials, and multiple easy-to-use output products. The inputs that are required to run the model include a digital elevation model for the terrain, specification of the dominant vegetation in the domain, and an input wind field. These inputs can all be downloaded from online sources using the *WindNinja* framework, if required. Three different options for definition of the initial wind field are possible: (1) A domain-average wind: an average value for the entire domain

at a defined height above the ground; (2) Multiple points: information about the wind at one or more observation points; (3) A coarser resolution wind field from a numerical weather prediction model. The code is open source and available on GitHub (*github.com/firelab/windninja*). It runs on both Windows and Linux operating systems. The core of the *WindNinja* framework are the two numerical solvers available to simulate the flow field, which both assume a neutrally-stratified flow with thermal parameterisations are available that allow some thermal effects to be approximated (e.g. non-neutral atmospheric stability and

diurnal slope winds) (Forthofer et al., 2009). The choice of solver (*COM* or *CFD*) is user-selectable at run time.The *COM* solver uses finite element techniques to minimally adjust (in a least-squares sense) the initial wind field to enforce conservation of mass. The governing finite element equations are solved using a conjugate gradient solver with Jacobi preconditioning on a terrain-following mesh with hexahedral cells that grow vertically with height.

In this work, the COM simulation was set up as described in Wagenbrenner et al. (2019), using *WindNinja version 3.5.3*. The

simulation was initialised with the domain-average initialisation option and the vegetation is specified as "grass", which sets the model roughness length to 0.01 m. The thermal parameterisations were not used. The computational mesh had a horizontal extent of 800 m long by 400 m wide by 26 m high (above sea level). The horizontal grid spacing was set to 4 m and the near-ground cell height to 0.1 m.

### 3.2   Model 2: *WindNinja-CFD*

*OpenFOAM version 2.2.0* is used for the CFD solver of *WindNinja* (Weller et al., 1998). Solutions to the steady-state, incompressible Reynolds-Averaged Navier-Stokes (RANS) equations are approximated using the *simpleFoam* solver, which is an implementation of the semi-implicit method for pressure-linked equations (SIMPLE). The standard k-epsilon model is used for turbulence closure. The governing equations are discretised using the finite volume method and solved on an unstructured mesh with mainly hexahedral cells, which follows the terrain. The inlet boundary conditions are defined as recommended in

Richards and Norris (2011).

The simulation in this work was set up as described in Wagenbrenner et al. (2019), and the domain-average initialisation option was used. The same set-up was used as for *WindNinja-COM*, except the CFD solver was chosen with the standard k-epsilon model for turbulence closure, and the computational mesh had dimensions of 800 m long by 400 m wide by 92 m high (above sea level), with a horizontal grid spacing of 3.8 m and a near-ground cell height of 3.8 m.

### 3.3   Model 3: *Zephy-CFD*

*ZephyCFD* is the CFD modelling chain of the wind resource assessment software *ZephyTOOLS*, which was developed by the company *ZephyScience* to combine all relevant wind resource assessment tools for a wind farm project on one platform. *Zephy-TOOLS* is a license-free software with highly parallelised calculations made possible by burst cloud-computing. *ZephyCFD* is





based on a three-dimensional (3D) RANS solver. The non-linear transport equations for mass, momentum and energy are thus solved considering a steady-state and isothermal incompressible fluid. The non-linear Reynolds stress tensor is modelled by the k-epsilon dual equation closure scheme, based upon coupled transport equations for the turbulent energy density and the turbulent dissipation rate. The meshing is done in a cylindrical domain to avoid re-meshing for each synoptic wind direction,

and uses an unstructured grid to focus on the points of interests without using an excessive amount of cells over the whole domain. The solver used is *OpenFOAM* with the SIMPLEC algorithm by default, but with the possibility for the user to change solver parameters. Boundary conditions are based on a log-law Atmospheric Boundary Layer (ABL) profile, for which the reference velocity, turbulent kinetic energy and inlet roughness can also be modified. Calculations are initialised on a coarser mesh generated automatically and then relaunched on the user-defined finer mesh. This allows for faster convergence times on

the final mesh and to check for grid-independence. Global convergence is based on the standard deviation of the wind shear results at the worst-converging point of interest for the last 50 iterations. *OpenFOAM* enables *ZephyCFD* to perform parallised calculations on the Cloud and to run all synoptic directions for a project simultaneously. The post-processing steps in *Zephy-CFD* allow results for wind resource assessments to be obtained, as well as energy yield analysis if power curves are provided. The spatial extrapolation of the wind properties is done by extrapolating wind measurements from met masts or lidars by the

speed-ups calculated in the CFD simulations.

As Bolund does not have the typical dimensions of a wind farm site, the user defined mesh properties in *ZephyCFD* were used for the meshing of the domain in this work. The overall domain was a cylinder of 3,800 m diameter and 1,000 m height, with cell resolutions varying horizontally from 2 m at the points of interest to 83 m at the boundary, and vertically from 0.2 m at ground elevation to 500 m at the top. Particularly high cell resolution was focused around the met masts and the areas identified

as having an average slope above a chosen threshold of 8°. In total the mesh was composed of 2.03 million cells. The input profile, defined at the boundary of the domain, was set as a log profile with user defined local roughness and reference velocity at reference height, as well as a uniform turbulent kinetic energy and a fixed kinematic viscosity. The values were chosen based on the profile given for the 270° case at the reference mast in the initial Bolund blind test (Bechmann et al., 2011).

### 3.4  Model 4: *Fluent-RANS*

*ANSYS Fluent* is a generic fluid dynamics tool for modelling the flow in industrial applications, and has capabilities for not only solving fluid flows, but also calculating turbulence, chemical reactions and heat flow. This ranges from furnace combustion, oil platforms, flow over aircraft wings and simulation of bubble columns. In this work, *Fluent* was first set up to solve the Reynolds-Averaged Navier-Stokes (RANS) equations. The RANS equations govern the transport of the averaged flow quantities, and can model the whole range of turbulence fluctuations using turbulence modelling, which approximate the turbulent flow. The

approach is very popular for engineering applications because it allows reasonably accurate modelling of a wide range of flow phenomena, but with significantly lower computational effort than other approaches such as Large Eddy Simulations (LES). In LES, the large-scale eddies are resolved explicitly in a time-dependent simulation using the filtered Navier-Stokes equations, reducing the error introduced by turbulence modelling. However, the filtered Navier-Stokes equations involve removing eddies smaller than the filter size (usually the mesh size), which does create additional unknowns that have to be approximated to





achieve closure. This method is thought to be more accurate than RANS, because the approximations only have to be applied to the very small scale eddies, which are less affected by the boundary conditions than large-scale eddies.

In this work, the meshing was done in *Fluent Meshing* using the new *Mosaic Technology*. This technique fills the bulk region with octree hexes but keeps a high-quality layered poly-prism mesh in the boundary layer and conformally connects these two
meshes with general polyhedral elements. The domain dimensions following a grid dependency study were 830 m by 450 m by 60 m. The mesh consisted of three regions: an outer region with a large mesh size (up to 15 m) and two refinement regions with a target mesh size of 0.5 m and 1 m. 15 boundary layer cells were used to capture near-wall effects, leading to a total cell count of about 10 million. The SST k-omega turbulence model was applied to attain closure.

### 3.5   Model 5: *Fluent-DES*

The Detached Eddy Simulation (DES) approach is a higher-fidelity approach than RANS, but lower than LES. It involves applying unsteady RANS in the boundary layer and LES treatment in the other flow regions. It effectively focuses the resolving of the turbulent eddies on the highly viscous boundary layer regions, significantly saving computational time compared to LES but still increasing the fidelity over RANS. It can therefore be seen as a compromise between RANS and LES.

In this work, the Delayed Detached Eddy Simulation (DDES) model was applied, which ensure that RANS is preserved
even in high-aspect ratio boundary layers. The same mesh and the same base settings were used as in the *Fluent-RANS* model described in the previous section, and the results from the RANS study were used to initialise the velocity fields and turbulence quantities. In contrast to the RANS simulation, the DES simulations were performed unsteadily with a time step of 50 ms. In order to introduce fluctuating velocities at the inlet, the *Fluent Synthetic Turbulence Generator* was used assuming the turbulence intensity given by the Bolund Hill blind test (Bechmann et al., 2011). After an additional unsteady initialisation
simulation, the wind speeds were averaged over 10 minutes. The SST k-omega turbulence model was applied to attain closure.

### 3.6   Model 6: *Palabos LBM/LES*

The Lattice Boltzmann Method (LBM) is an alternative type of CFD for fluid simulation. Instead of solving the Navier–Stokes equations directly, a fluid density on a lattice is simulated with streaming and collision (relaxation) processes. It uses the immersed boundary method and therefore does not require a geometry-conforming grid. *Palabos* is a LBM code developed at
the University of Geneva, which has a straightforward programming interface and allows fluid flow simulations to be set up or the library to be extended with new models quite easily. The native programming interface in written in C++, and the library has hardly any external dependencies (only Posix and MPI), making it easy to deploy on different platforms. However, the lack of graphical user interface means that the tool takes some getting used to and requires programming in order to get run simulations. Palabos can be downloaded and used for free under the terms of an open-source AGPLv3 license.
The generation of the orthogonal mesh is fully automated in *Palabos*. The surface mesh is provided in stereo-lithography format (STL) and *Palabos* then converts the surface description of the domain into a volume description. In this work, the mesh was created with a constant cell size of 0.5 m, resulting in a total cell count of approximately 40 million nodes. Three different mesh sizes were tested in a separate study (Schubiger et al., In Review). The dimensions of the most effective mesh





chosen for this study was 40 m by 250 m by 525 m. The domain boundaries consisted of Bounce Back nodes. As there is no way to describe different roughness lengths in *Palabos* yet, the water surface was modelled with a zero gradient for tangential velocity components (free-slip). In addition, a simple velocity fluctuation model was implemented by synthetically creating the fluctuations based on the value of the turbulence intensity of the Bolund blind test, in order to generate turbulence at the inlet. A

standard *Smagorinksy Sub-Grid Scale* model was used to capture sub-grid-scale motions. The *Palabos* framework offers more sophisticated LES and boundary models; however, the basic models were used here for simplicity. The possibilities of these advanced models promises even more accurate solutions then obtained with this simple approach. More details of the Palabos simulations can be found in Schubiger et al. (In Review).

## 4    Results

### 4.1    Definition of test case

As mentioned in the previous section, the prescribed flow field from the Bolund Hill blind test experiment was used in this work. Only one wind direction was chosen for simplicity, because the focus was on the development of the parameters for the comparison metrics. The 270º wind direction was chosen in this case due to the high availability of simulation data in this wind direction.

### 4.2    Parameter definition

#### 4.2.1    Scores before carrying out simulations (predicted scores)

The *skill* score of each model can be estimated **before** running the simulations by considering the parameters that are expected to affect the accuracy of the results. These were divided into factors regarding the *model* (mathematical model, time step, simulation length, grid quality, degree of turbulence and terrain approximation), the *input data* (terrain complexity, surface

roughness complexity, atmospheric stability, quality of measurement data, quality of terrain data, quality of surface data and quality of atmospheric data) and *other parameters* (skill of user, robustness of model and accuracy of validation). These factors were quantified for each model by providing the modellers with a tabular template in which pre-defined lower and upper limits were given, and then converting these to percentage values assuming linear behaviour between the limits. The lower and upper limits were pre-defined by the authors, and are expected to be improved and tuned as the project progresses and more data is

collected. The *input data* is included in this quantification in order to allow comparisons between different sites on the same plot in the future, although in this work, the values remained the same for each model.

In order to demonstrate the type of parameters and the quantification method used, details are shown in Table 1 for the *model* parameters only. This shows, for example, that grid quality is scored by asking the questions "How well are the software recommendations regarding orthogonal quality, skewness and aspect ratio fulfilled?", "Has grid independency been proven via

a grid study?" and "Are the estimated y+ values within the software recommendations of the applied boundary layer model?". A full list can be found in the publicly-available template (Barber, 2019).



**Table 1.** Parameters for estimating the skill score before running simulations - *model* parameters only.

| Parameter | Value(s) to quantify | Name | Quantification method |
|---|---|---|---|
| Model | | | |
| Mathematical model | Underlying aerodynamic equations | $M_A$ | Estimate using pre-defined values or linear interpolation between them |
| | Underlying thermodynamic equations | $M_{TH}$ | Estimate using pre-defined values or linear interpolation between them |
| Time step | Size of time step (for unsteady simulations) | $M_T$ | Estimate using pre-defined values or linear interpolation between them |
| Simulation length | Length of simulation period (for unsteady simulations) | $M_L$ | Estimate using pre-defined values or linear interpolation between them |
| Grid quality | Grid quality | $M_{GQ}$ | How well are the software recommendations regarding orthogonal quality, skewness and aspect ratio fulfilled? |
| | Quality of grid independency | $M_{GI}$ | Has grid independency been proven via a grid study? |
| | Boundary layer resolution | $M_{GB}$ | Are the estimated y+ values within the software recommendations of the applied boundary layer model? |
| Degree of turbulence | Reynolds number | $M_R$ | Estimate approximate Re to check if laminar or turbulent (if relevant), calculate based on flow velocity and distance to met mast of interest |
| Terrain approximation | Ability of 3D grid to adapt to terrain | $M_{TA}$ | How well can the grid adapt to the terrain (visually check that the geometry is properly captured)? |

Next, each *skill* score parameter was given a weighting according to the expected relative impact of the parameter on the overall *skill* score. Again, these values were defined by the authors and are expected to be improved during the course of the project. The weightings for *skill* score **before** the simulations used in this study are shown in Fig. 4. It can be seen that a large weighting is given to the quality of the aerodynamic model; however, other important parameters include the average deviation

5  from the measurements of a validation study (if existing), as well as other parameters such as the size of the time step, the grid quality and the Reynolds number. The remainder of the values can be found in the template (Barber, 2019).

The *skill* scores assigned to each parameter for each model were multiplied by the weightings and divided by 100 to give a percentage weighted score. All these percentage weighted scores were then added together and divided by the sum of the weightings to give an overall score as a percentage. In order to demonstrate the pre-defined limits and the assigned scores and weightings, details are shown in Table 2 for the *model* parameters only, with scores given for Model 6 as an example. It

10  can be seen that this model has a very high weighted score for the parameter defining the underlying aerodynamic equations, $M_A$, due to the fact that it uses LES. Other high scores include the below 1 s time step size and the high level of fulfillment





**Table 2.** Quantification of parameters for estimating the skill score before running simulations - example for Model 6.

| Name | Pre-defined values | Absolute score | Score in % | Weighting in % | Weighted score |
|---|---|---|---|---|---|
| Model | | | | | |
| $M_A$ | Linear or conservation of mass model = 2%, RANS = 40-60% depending on numerical model, LES = 90% | LES | 90 | 100 | 90 |
| $M_{TH}$ | None = 0%, temperature modelling = 40%, coriolis force = 40%, both = 80% | None | 0 | 0 | 0 |
| $M_T$ | 1 hour = 10%, 10 minutes = 50%, 1 minute = 80%, 1 second = 100% (steady-state = 100%) | 1 s | 100 | 20 | 20 |
| $M_L$ | 1 minute = 20%, 10 minutes = 40%, 1 hour = 60%, 1 day = 100% (steady-state = 100%) | 10 min | 40 | 20 | 8 |
| $M_{GQ}$ | Not at all = 0%, partly = 40%, mostly = 60%, fully = 100% | Fully | 100 | 20 | 20 |
| $M_{GI}$ | No = 0%, yes with minor problems = 50%, yes = 100% | Partly | 20 | 20 | 4 |
| $M_{GB}$ | Not at all = 0%, partly = 40% mostly = 60%, fully = 100%, not relevant = 100% | Partly | 10 | 20 | 10 |
| $M_R$ | Above 100,000 / laminar = 0%, below 100,000 / turbulent = 100%, not relevant = 100% | Turbulent | 100 | 10 | 10 |
| $M_{TA}$ | Not at all = 0%, partly = 40% mostly = 60%, fully = 100%, not relevant = 100% | Mostly | 60 | 10 | 6 |

of the software recommendations regarding orthogonal quality, skewness and aspect ratio of the grid. Particularly low scores include the fact that grid independency has not been fully proven via a grid study (different grid dimensions were not tested for this model, see Schubiger et al. (In Review) for more details), that the estimated y+ values are only partly within the software recommendations of the applied boundary layer model, and that the user is not experienced with the software (value not shown in Table 2).

The *cost* score of each model can be estimated **before** running the simulations by considering the factors that are expected to affect the costs. As shown in Table 3, these consist of investment and staff costs related to the software (per project), the time to learn and training costs (per project), the expected simulation set-up effort, the cost of the expected simulation run-time and the expected post-processing effort. In order to be able to compare the results between projects, the same staff costs ($80/hour) and the same number of projects per year (12) are assumed for each model. The absolute cost values for each model were added up and the total was scaled linearly between $0 (0%) and the most expensive model in the study (100%). The absolute scores for Model 6 are shown in Table 3 as an example (the model with the highest predicted *cost* score). Although the software is





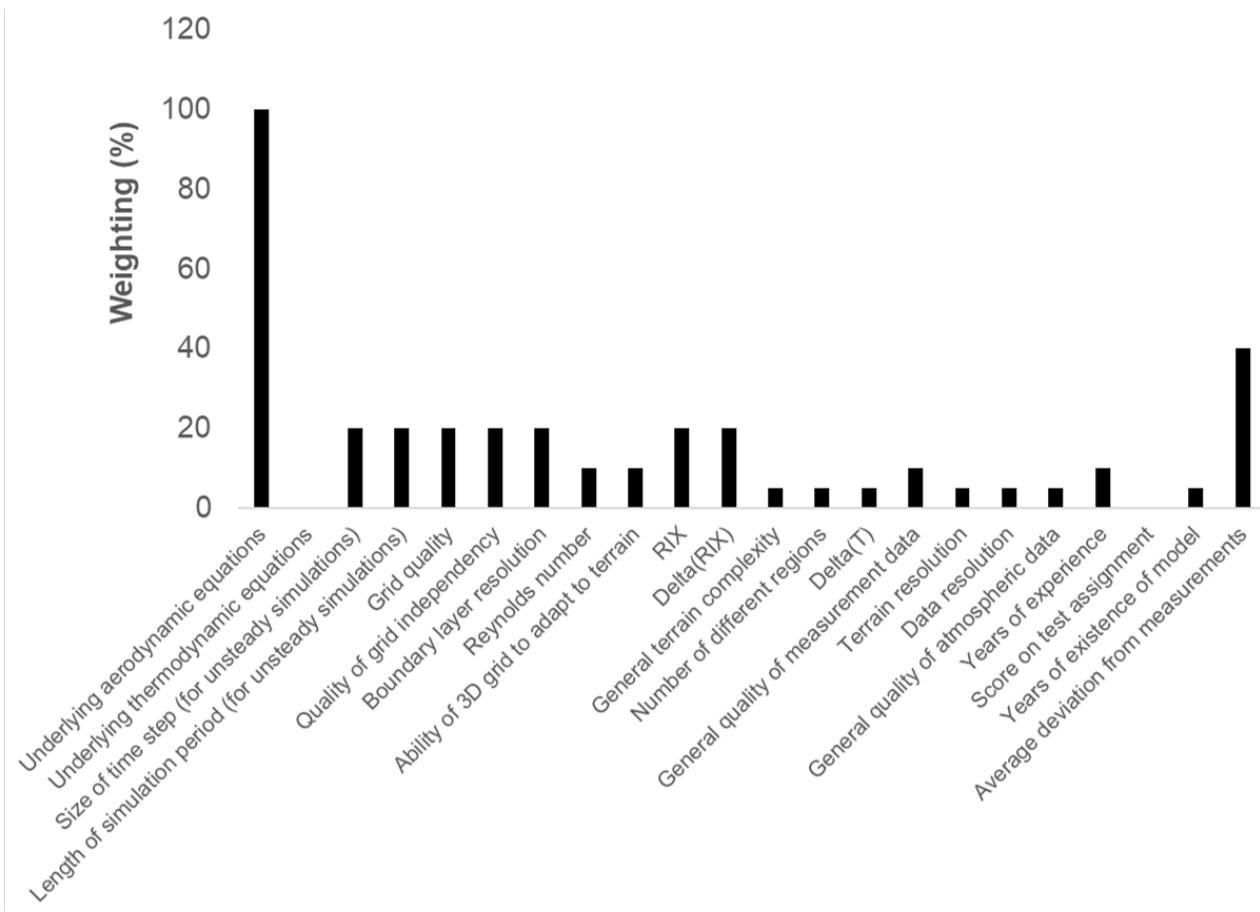

**Figure 4.** Weightings assigned to skill score parameters in this study.

free, the expected simulation set-up time is very high, because the interface is not designed specifically for this wind energy application and a large amount of work is required to set the simulations up. This is expected to decrease the more experience the user has with the software.

### 4.2.2 Scores after carrying out simulations (actual scores)

5 The *skill* score of each model can be calculated **after** running the simulations by comparing simulated data to measured data for the same input conditions. Many factors related to wind modelling are important for the eventual Annual Energy Production (AEP) calculation; however the four key values for wind energy wind modelling applications have been identified as (1) Wind velocity magnitude; (2) Wind direction (or all three components of wind velocity); (3) Turbulence intensity; (4) Shear factor. In this work, only the wind velocity magnitude has been considered for quantifying the *skill* score **after** running the simulations.

10 The simulated wind speed magnitudes for four measurement positions at met masts along the 270° line from Fig. 2 are plotted against the measured values for each model in Fig. 5, with the coefficient of determination, $R^2$, marked on each plot.





**Table 3.** Parameters for estimating the costs before running simulations, example for Model 6

| Parameter | Value(s) to quantify | Name | Absolute score ($) |
|---|---|---|---|
| Software costs | Investment costs plus support/license costs per year, divided by number of projects | $C_I$ | 0 |
| Time to learn and training costs | Staff costs invested in learning x hourly rate + training costs, divided by total number of projects carried out by staff member | $C_T$ | (15 days x 8.4 hr./day x $80/hr.)/(12 projects per year) = $840 |
| Expected simulation set-up effort | Staff costs to set up simulations (per project) | $C_S$ | 6 days x 8.4 hr./day x $80/hr. = $4,032 |
| Expected simulation run-time | Costs for running one simulation (use the final set-up that creates the results entered in this table) | $C_R$ | 4 days x 8.4 hr./day x $0.04/core x 120 cores. = $161 |
| Expected post-processing effort | Staff costs to post process results (per project) | $C_{PP}$ | 3 days x 8.4 hr./day x $80/hr.= $2,016 |

These four points were chosen in order to ensure comparability, as they were the only four points for which all the measurement results were available. In general, the models all perform reasonably. The details of the flow are examined further in Schubiger et al. (In Review) for Model 4, Model 5 and Model 6. To calculate the *skill* score for each model, the Root Mean Square Error between the simulations and the measurements at these four measurement positions was calculated for the wind velocity magnitude. These values were then scaled linearly to a percentage value using upper limits of 3 m/s, and a lower limit of zero. The upper limit resulted in an absolute score of 0%, as a high error results in a low *skill* score.

The actual *cost* score of each model **after** running the simulations was defined by recording the actual investment and staff costs related to the software, the time to learn and training costs, the simulation set-up effort, the cost of the simulation run-time and the post-processing effort. As for the *cost* score **before**, the absolute costs for each model were added up and the total was scaled linearly between $0 (0%) and the most expensive model in the study (100%).

## 4.3 Resulting scores

After definition of these parameters, their values were collected for each tool by providing the modellers with a tabular template containing detailed instructions and descriptions of each parameter (Barber, 2019). In this template, full descriptions of all the parameters as well as their upper and lower limits can be found. Although this process involved iterative improvements following inputs from the participants, it remained a blind test and the modellers were not able to tune their results to improve them.




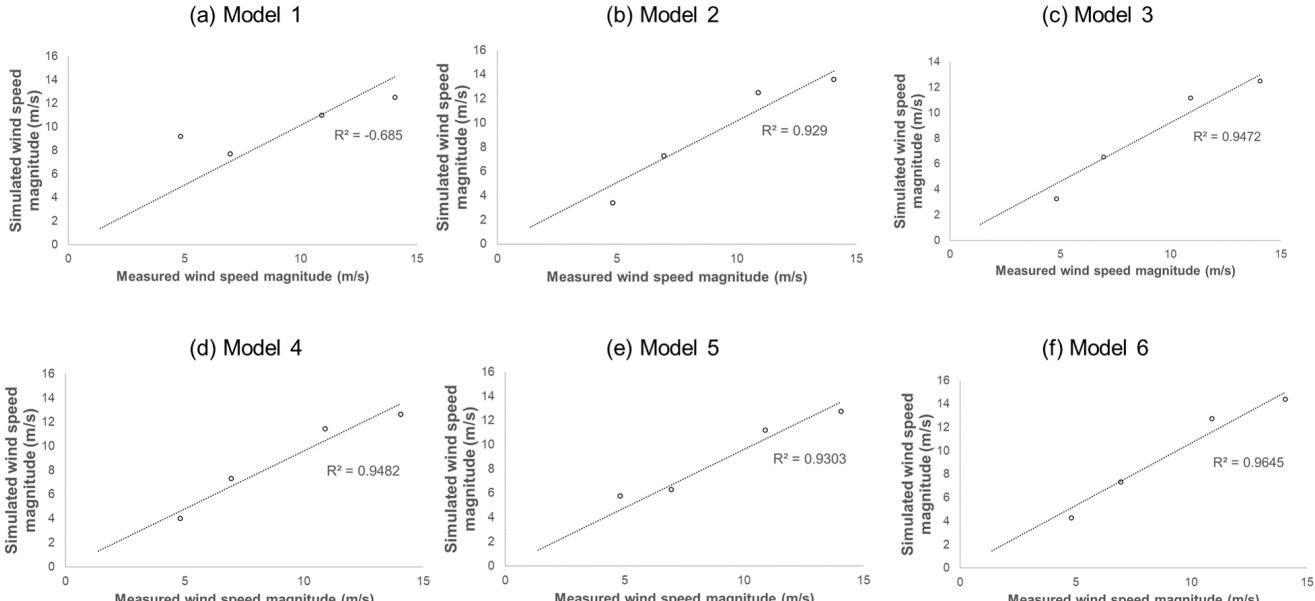

**Figure 5.** Correlations of measured and simulated wind speeds for each model for four measurement points.

The resulting *skill* and *cost* scores for all the models, both **before** and **after** the simulations, are shown in Fig. 6. It can be seen that Model 2 would probably be chosen as the the most cost-effective model, both **before** and **after** carrying out the simulations (given by the flattening-off point of the curve). This shows that the current method is successfully able to help modellers choose the most effective model for this test case, without having to carry out the simulations before deciding.

Examining first the predicted results **before** the simulations were carried out (the black diamonds), the expected shape introduced in Fig. 1 can be seen in Fig. 6, with each model number marked. There is an initial increase in *cost* and *skill* scores, followed by a flattening-off. Model 1 is very quick and easy to apply, giving it a low expected *cost* score, but is expected to be the least accurate model due to its simple conservation of mass approach, giving it a low expected *skill* score. It is mainly the highly-weighted parameter *aerodynamic equations* that is responsible for the very low *skill* score, and the short set-up and post-processing time that results in the low *cost* score. Models 2 and 3 are expected to perform very similarly, because they both apply the CFD RANS method and both involve a similar user-friendly, application-specific set-up, with higher expected *skill* and *cost* scores than Model 1. Model 4 is expected to have a similar *skill* score to Model 2 and 3, due to the same CFD RANS method. However, this model is a general tool for industrial applications and has not been designed specifically for this application. Therefore the set-up and post-processing times are expected to be much higher, giving it a higher expected *cost* score than Model 2 and Model 3. Model 5 is expected to have a slightly higher *skill* score than Model 4, due to the DES approach; however, the run-time is expected to be much higher and therefore the expected *cost* score is significantly higher. Although Model 6 is using a more accurate LES model and the expected *skill* score is very high, its more efficient LBM





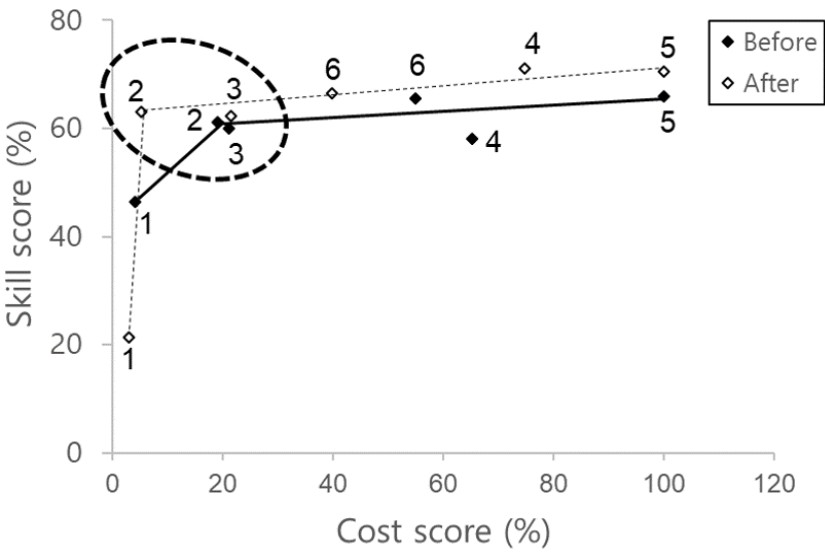

**Figure 6.** Results of skill and cost scores for all models before and after the simulations.

approach results in a lower expected *cost* score than Model 4 and Model 5. However, this time saving is expected to be partially offset by the high simulation set-up time due to the fact that the software is not designed specifically for this application and is new to the user in this case, and therefore the expected *cost* score is not as low as the RANS models. The *cost* score of Model 6 does, however, have the potential to be reduced significantly by the development of an application-specific module in 5 the future, or after gaining more experience with the model. In general, it is interesting to note how low the importance of the simulation run time is, compared to the other costs. This is examined further in Section 4.5.

Now looking at the calculated results **after** the simulations were carried out (hollow diamonds), the same overall shape can be seen very clearly, although not all the absolute values are the same as the predicted values. As for the predicted scores, Model 2 would probably be chosen as the most cost-effective model. The closer together the numbers are on Fig. 6 the better 10 the prediction of the scores. Some of the values have been predicted very well, such as the *cost* and *skill* scores of Model 3, the *skill* score of Model 2 and the *cost* scores of Model 1 and Model 5. Others have not been predicted as well, including the *skill* scores of Model 1 and Model 4 as well as the *cost* scores of Model 4 and Model 6. A direct comparison of the *cost* and *skill* scores before and after the simulations is shown in Fig. 7. It can be seen that the prediction of the *cost* scores is much better than the prediction of the *skill* scores in general. This is due to the complex nature of the *skill* score prediction method, 15 which requires further investigation with far more data points. This is currently being done via a public simulation challenge in collaboration with IEA Wind Task 31 (*www.iet.hsr.ch/windenergy*).





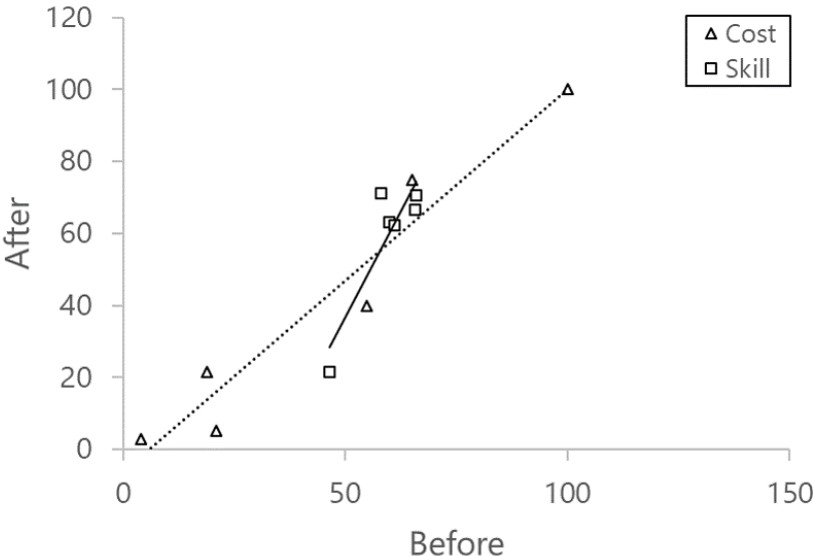

**Figure 7.** Direct comparison of skill and cost scores before and after the simulations.

## 4.4 Choice of most cost-effective model

As mentioned in the previous section, Model 2 has been identified as the most cost-effective tool for this application, both **before** and **after** carrying out the simulations. This shows that this novel method of predicting the *skill* and *cost* scores of a range of models for a given project in order to help modellers choose the best model for their needs works effectively. Model

2 is a RANS CFD model developed specifically for wind modelling applications, and has the most suitable combination of modelling accuracy and quick set-up time for this application. Model 3 would also be a very good choice for this application, reaching good accuracy and reasonably low costs. It is important to note that this choice of the most cost-effective model is highly dependent on the type of problem being solved. For example, for projects in which the detailed simulation of flow separation and thermal effects are important, the expected improvement in *skill* score of LES models compared to RANS

CFD models could very possibly outweigh the *cost* score savings of the RANS models. Additionally, the choice of most cost-effective model is highly dependent upon the experience of individual modellers as well as on the availability of validation and grid independency studies. This method should therefore be applied by the modeller for each individual wind energy project. Furthermore, the modeller may need to apply further constraints that could be added to the plot of *skill* against *cost* score as in Fig. 1, which may then affect the decision process. Further work is being carried out on the suitability of this new method for

other project types, such as terrain complexity and atmospheric stability, as well as for calculating all wind directions and the AEP and including mesoscale nesting or forcing. A large number of inputs will be collected as part of a simulation challenge in collaboration with IEA Wind Task 31.





### 4.5 Further analysis

In order to further examine this new method, the scores predicted in this work were compared to a simplified prediction method using only the parameter *aerodynamic equations* for the *skill* score and only the relative run time costs for the *cost* score. The results are shown in Fig. 8, which compares *skill* against *cost* scores **before** carrying out the simulations for the new (current) method and for the simplified method for each model. With the simplified method (hollow triangles), it is not possible to find a pattern in the results or choose the most cost-effective model. This highlights the danger of predicting the *skill* score of a model only using simplified assumptions such as the accuracy of the aerodynamic equations and quantifying the *cost* score only using the run time. There are far more other parameters that are important for assessing the *skill* and *cost* scores of a model.

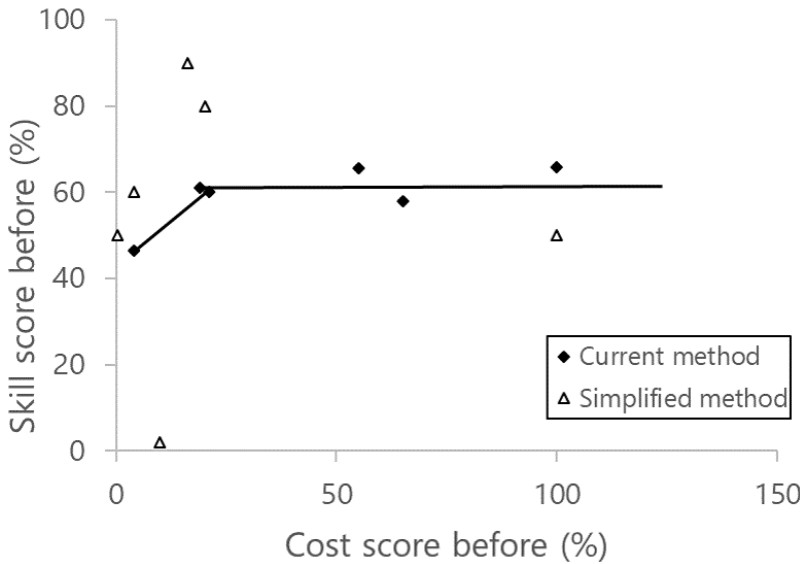

**Figure 8.** Comparison of prediction before the simulations for the current method and for a simplified method only using the aerodynamic equations run time.

For example, the distribution of the actual costs for each model are shown in Fig. 9. This shows that the simulation run time costs (shown in light grey) are tiny in comparison to the simulation set-up and post-processing effort for all of the models. Additionally, the relative expense of the *Fluent* software in terms of the license costs as well as the set-up time are clear to see (Model 4 and Model 5).

Finally, in order to obtain a further understanding for how the results may change on altering the weightings of the parameters, a sensitivity study was carried out, in which the weightings of the four most strongly weighted *skill* score parameters were altered systematically in order to give a total of three weightings per parameter. The effect of these weightings are shown in Fig. 10. The magnitude and direction of the change in the parameter due to the change in weighting is dependent on the value of the score given to each parameter for each model as well as on the relative importance of the parameter to the overall score,



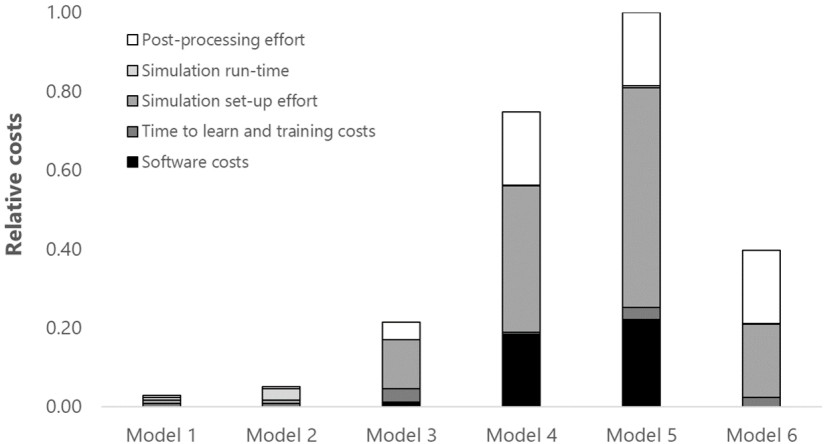

**Figure 9.** Distribution of the relative costs for each model.

and therefore is different for each parameter and model. For example, for the parameter *aerodynamic equations*, reducing the weighting increases the *skill* scores of Model 1, 2 and 3, but decreases the *skill* scores of Model 4, 5 and 6. This is because the score given to Model 1, 2 and 3 is relatively low, and therefore reducing the weighting increases the relative importance of the other parameters and increases the overall score, which represents an average of all weighted scores. As the score given to

Model 4, 5 and 6 is relatively high (above 50%), reducing the weighting of this parameter decreases the overall *skill* score as its importance reduces. It can also be seen that the magnitude of the change in *skill* score is much higher for Model 1 than for the other models. This is because the lower the weighting, the less the very low score causes an overall score reduction, and the higher the score becomes. Furthermore, as expected, changes to the weighting of the parameter *aerodynamic equations* also have a larger effect on the overall shape of the graph as the other parameters do. Further work is undergoing on the refinement

of this weighting procedure.

It should be further noted that this new method has a high potential to be extended to a wide range of other research and industry simulation applications, including in the automotive and aerospace areas, for which many different tools are available and the choice of the most cost-effective tool is also highly challenging.

## 5   Conclusions

In this work, a new method for helping wind modellers choose the most cost-effective model for a given project was developed by applying six different Computational Fluid Dynamics tools to simulate the Bolund Hill experiment and studying appropriate comparison metrics in detail.

This was done by firstly defining various parameters for predicting the *skill* and *cost* scores **before** carrying out the simulations as well as for calculating *skill* and *cost* scores **after** carrying out the simulations. Weightings were then defined for

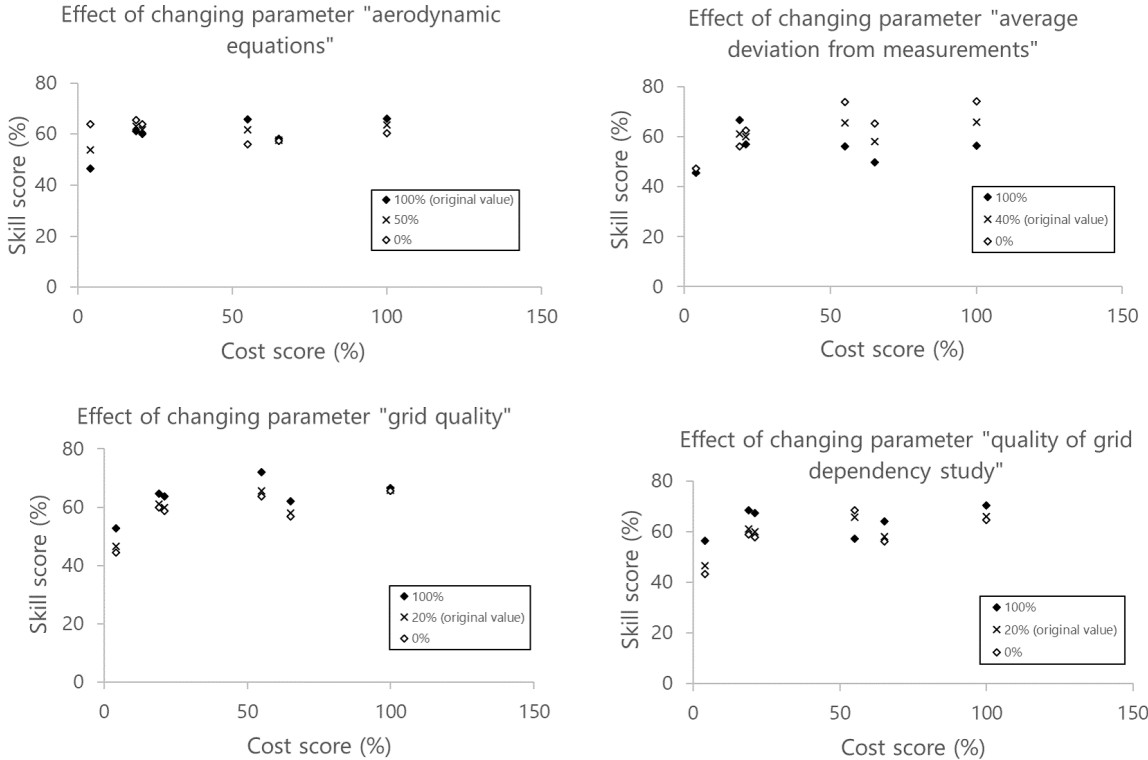

**Figure 10.** Effect of changing weightings of the four most strongly-weighted parameters on the skill score vs. cost score curve.

these parameters, and values assigned to them for the six tools using a template containing pre-defined limits in a blind test. An iterative improvement process was applied by collecting inputs from the participants of the study. This allowed a graph of predicted *skill* score against *cost* score to be produced, enabling modellers to choose the most cost-effective model without having to carry out the simulations beforehand. The most effective model is the one with the highest *skill* score for the lowest

5  *cost* score, at the flattening-off part of the curve.

The results showed that this new method is successful, and that it is generally possible to apply it in order to choose the most appropriate model for a given project in advance. This was demonstrated by the good match between the shapes of the *skill* score against *cost* score curves **before** and **after** the simulations, and by the fact that the tool at the flattening-out point of the curve is the same **before** and **after** carrying out the simulations.

10  It was also shown how important it is to take into account other factors that may affect the accuracy and costs of a wind modelling simulation as well as the quality of the aerodynamic equations and the run-time.

Several improvements to the method are being worked on, by further examining the discrepancies between the predicted and actual *cost* and *skill* scores. Additionally, the method is being extended for calculating all wind directions and the Annual



Energy Production, as well as to include mesoscale nesting or forcing. A large number of inputs are being collected as part of a simulation challenge in collaboration with IEA Wind Task 31.

The method has a high potential to be extended to a wide range of other simulation applications.



*Data availability.* The tabular template is available at http://doi.org/10.23728/b2share.b5650658062047189649c8755dec79fc

*Author contributions.* The contribution of the authors in this paper is:

– Sarah Barber: design of the method, acquisition and management of project, coordination of manuscript.

– Alain Schubiger: Fluent and Palabos simulations and descriptions, submission of parameter scores for Fluent and Palabos, iterative
5    improvement of template.

– Natalie Wagenbrenner: WindNinja simulations, WindNinja description, submission of parameter scores for WindNinja, iterative improvement of template.

– Nicolas Fatras: Zephy simulations, Zephy description, submission of parameter scores for Zephy, iterative improvement of template.

– Henrik Nordborg: supervision of Sarah Barber.

10   *Acknowledgements.* -

    Thanks to the funders of this project: the Swiss Federal Office of Energy (contract number SI/501807-01), the German Federal Environmental Foundation as well as the companies Enercon, ewz, ADEV and BKW.





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
