# Peer review of "A new method for the pragmatic choice of wind models for Wind Resource Assessment in complex terrain"

_Wind Energy Science, 2019_

## Referee Comment (RC1) · Javier Sanz Rodrigo (Referee) · 7 Feb 2020

Interesting article discussing the inherent challenge of the wind industry when it comes to deciding about the most suitable model to predict the wind resource at a given site with varying degrees of flow and terrain complexity. The objective is to present a decision making tool that does not require running potentially costly simulations before a decision is made. To design this tool, data is collected from a benchmark study around the Bolund hill case. While the paper describes the challenge and many factors affecting model cost and skill, the amount of data collected and the subjectivity of some of the criteria makes the conclusions about the effectiveness of the method questionable.

[Figure]

I would postpone the publication in a scientific journal until the authors have collected more data to build a complete case-study that support the conclusions.

The main limitation in this type of studies is the representativeness of the conclusions when only one site has been evaluated. The whole process depends on an iterative process to find out the best weightings on the criteria. The resulting decision making tool should be applied to independent data from other sites to make sure you are not overfitting to the particular characteristics of the Bolund case and the models and modellers inolved in the study. While this is being addressed in a follow-up study in the IEA Task 31, I believe it is too early to claim that the methodology is demonstrated.

Another important aspect is that the methodology does not allow to separate the influence of the user in skill and cost. User and model experience are combined which makes the whole scoring process very subjective to the interpretation of the user. The relationship found is a bit artificial when you are mixing general purpose and research codes with application-specific tools. I think that industry will automatically discard general purpose and research codes simply because they don't have enough credibility or they can't be used in an operational setting. If you leave out models 4, 5 and 6, I'm not sure if the relationship between before and after holds.

At a benchmark level, I would not extrapolate the results to an operational environment because accuracy is quantified in terms of just a few flow cases here while wind resource applications require integration to the full wind climate to predict AEP. Here, skill and cost are highly influenced by the methodology that is applied to combine microscale flow simulations with the measured or simulated wind climate. This is mentioned in the paper as future work but I'm afraid this is necessary to present a complete case study that addresses the challenge presented in the paper.

Other remarks

1.18 Abstract: "Several ..." I would leave out this last part where you explain about ongoing work.This belongs to the conclusions/discussion section of the paper.

4.1: "These metrics can only be used if the number of data points is high enough to allow statis-tical analysis. For wind energy applications, ten-minute averages are usually sufficient, and therefore these metrics are not necessarily applicable here". This sentence is not appropriate. The validation data points that are object of analysis in the COST-732 guidelines are related to sensor locations in field and laboratory experiments. Validation on many 10-min samples is not equivalent if those samples are all based on the same sensor. In wind energy we more often have low number of samples (sensors and sites) to cover the validation range of operational wind turbines. Therefore, the metrics are still valid and it is more often the case that we lack the statistical significance in our validation campaigns.

Section 2.1: I would add Hills et al (2015) to the list of V&V frameworks since it specifically addresses wind energy models Hills R., Maniaci D., Naughton J. (2015) V&V Framework. Sandia Report SAND2015–7455, September 2015 http://prod.sandia.gov/techlib/access-control.cgi/2015/157455.pdf

4.23: "Also as part of IEA Wind Task 31, a Wind Energy Model Evaluation Protocol (WEMEP) has been developed (Rodrigo, 2019)" WEMEP is still at an early stage of development so I would be cautious referencing this. Maybe this belongs to the discussion part where you put this work in a wider context linked to the IEA Task 31 objectives.

5. Figure2: Add citation to Benchmann et al (2011) and make sure you are allowed to copy the image here.

Section 3. Models. The description of the models is too superficial, more related to the code framework than the actual CFD models. Please add a summary table where you can compare the simulations from a computational (cost related) and physics (skill related) point of view. At least you should include the mesh size, spatial and temporal resolutions, wall time, boundary conditions, turbulence parameterization, etc.

8. Fluent-RANS/DES: Using default settings of a generic CFD solver to solve atmospheric boundary layer flow is problematic. The authors do not mention any specific settings of the code to deal with this particular application other than the grid dimensions and choice of turbulence model. Switching from RANS to DES will not necessarily make the results more reliable if the boundary conditions or turbulence coefficients are not adequate for ABL flows in the first place.

11. Table 1. The definition of Re based on the distance to the met mast is very arbitrary. Shouldn't it be based on the hight of the hill? Maybe I would use a fixed definition of Re and then ask the modeller how much they deviate from it in the simulation. Still, the relevance of Re in this study should be very low.

12. Table 1. Grid quality is something that cannot be given for granted even if the user manual of the software explains about the good qualities of the generated mesh. This is application and case specific. Unfortunately, to rank high on this factor you will make the model more costly. Quantifying this should have been part of the benchmark.

12. Table 2./Figure 4. These criteria seem to assume that all the codes have been validated for the purpose of the benchmark and then the skill/score depends on the computational cost and the years of experience of the user and the code. In reality I would expect better performance from a linearized model that has been calibrated for the application than a generic LES model that is used out-of-the-box. Maybe this is all covered in the "years of experience of the model" but I also don't see how time matters when you could have used a more specific metric like "number of published validation studies on wind resource assessment" which addresses both the maturity of the model and the experience of the user being aware of the validation track record.

---

## Referee Comment (RC2) · Anonymous Referee #2 · 19 Feb 2020

This manuscript describes a new method to support the choice of wind models for wind resource assessment. Several parameters of model skill score and cost are taken into account. Although the method is an interesting new approach, it definitely needs further testing and evaluation. At this stage, the new method has only been tested for the Bolund Hill experiment for a single case study; as a test quantity the wind velocity is chosen. I think that the new approach needs further evaluation, especially also for the other relevant variables in wind resource assessment.

[Figure]

**General comments**

1. **Skill weighting.** The authors take many factors into account for the weightings assigned to the skill score. However, it is not entirely clear from the text how the weights of the individual contributions were determined. The aerodynamic solver was chosen to be the most important factor. Why? Is this general knowledge (are references of publications available)?
   Were these factors chosen with a specific focus on the well-known Bolund Hill experiment? However, if this skill score framework is meant to be applied to other test sites or case studies, sparse input data (e.g., insufficient terrain data, coarse atmospheric profiles, or low-quality observations) pose a challenge to a model, even before the aerodynamic solver can have any effect on the model skill. I wonder why the input data quality is given so little weighting.

2. **Choice of the wind velocity**. The authors mention the four key values for wind energy assesment (wind velocity, wind direction, turbulence intensity, and the shear factor), but only show the results for the wind velocity in the rest of the manuscript. Since the Bolund Hill experiment has a rich observational dataset, I am wondering why the authors omit the other three quantities? I guess that at least the wind direction, but also the turbulence intensity can be extracted from the observations. The manuscript's title includes "complex terrain", and since the turbulence structure over such a surface is known to be complex, an assesment of the turbulence and the shear factor with the new skill score framework would definitely enrich the manuscript.

3. **Generalization.** The authors prove a new and interesting cost versus skill estimation model. However, after reading the manuscript, the question would be how easy it is applicable for other test sites or situations. The authors mention that testing for other wind directions and other atmospheric stabilities has to be carried out and will be underway. However, the applicability besides the Bolund

Hill test site is still somewhat limited and definitely needs further testing. For this manuscript, it might be sufficient to add the other three relevant quantities (wind direction, turbulence intensity, and the shear factor) to gain more insight on the new method, but the authors need to stress in their conclusions more precisely that this is novel, preliminary work which still needs (skill parameter) adjustment.

**Specific comments**

**Section 3:** The authors describe every chosen model, but it would be useful to summarize the model's chosen settings (such as grid spacing, aerodynamic solver, turbulence treatment,...) in a accompanying table. Furthermore, references to the model code and descriptions are missing for some models (see technical corrections below).

**Technical corrections**

- page 5, line 7 (and follow-up occasions): Put the two references in a shared bracket

- page 7, line 10: What does the abbrevation COM stand for? Conservation of Mass?

- page 7-8: Can you provide a published reference to the *ZephyCFD* modelling chain?

- page 8: Can you provide a published reference to the *ANSYS Fluent* tool?

- page 8: Can you provide a published reference to the *Palabos* code?

- page 13, line 7: Repetition: "[...] however the four key values for wind energy wind modelling applications have been identified as [...]"

- General remark to the references: for some journals, the authors use the abbreviations, for some they don't, and at some references the dois are missing. Make sure that the references follow WES' guidelines.

- page 23, line 29: Typo in the link. It should say https://wemep.readthedocs.io/en/latest/mep/mep.html

- page 23, line 30: You might add the doi to this reference.